# Yoga Pose Estimation Using Angle-Based Feature Extraction

**DOI:** 10.3390/healthcare11243133

**Published:** 2023-12-09

**Authors:** Debanjan Borthakur, Arindam Paul, Dev Kapil, Manob Jyoti Saikia

**Affiliations:** 1Department of Psychology, University of Toronto, Toronto, ON M5S 3G3, Canada; 2Independent Researcher, Malden, MA 02148, USA; 3Doctor On Click, 247A/247B Victoria Street, Singapore 188033, Singapore; dev@onewellness.com.sg; 4Department of Electrical Engineering, University of North Florida, Jacksonville, FL 32224, USA

**Keywords:** yoga, computer vision, machine learning, pose estimation

## Abstract

Objective: This research addresses the challenges of maintaining proper yoga postures, an issue that has been exacerbated by the COVID-19 pandemic and the subsequent shift to virtual platforms for yoga instruction. This research aims to develop a mechanism for detecting correct yoga poses and providing real-time feedback through the application of computer vision and machine learning (ML) techniques. Methods and Procedures: This study utilized computer vision-based pose estimation methods to extract features and calculate yoga pose angles. A variety of models, including extremely randomized trees, logistic regression, random forest, gradient boosting, extreme gradient boosting, and deep neural networks, were trained and tested to classify yoga poses. Our study employed the Yoga-82 dataset, consisting of many yoga pose images downloaded from the web. Results: The results of this study show that the extremely randomized trees model outperformed the other models, achieving the highest prediction accuracy of 91% on the test dataset and 92% in a fivefold cross-validation experiment. Other models like random forest, gradient boosting, extreme gradient boosting, and deep neural networks achieved accuracies of 90%, 89%, 90%, and 85%, respectively, while logistic regression underperformed, having the lowest accuracy. Conclusion: This research concludes that the extremely randomized trees model presents superior predictive power for yoga pose recognition. This suggests a valuable avenue for future exploration in this domain. Moreover, the approach has significant potential for implementation on low-powered smartphones with minimal latency, thereby enabling real-time feedback for users practicing yoga at home.

## 1. Introduction

The field of computer vision focuses on developing algorithms and systems for the automatic processing and analysis of visual data. Among its applications are image and video recognition, object detection, and human pose estimation. By analyzing images or videos of people performing yoga poses, computer vision techniques can help identify the specific pose being performed by an individual. The importance of pose estimation in computer vision cannot be overstated. Pose estimation is the process of estimating the configuration of a human body (pose) from a single (typically monocular) image [1]. This is a challenging problem as it depends on the resolution of the image, the background clutter of the surroundings, and other noise. Pose estimation has become essential in the field of sports, fitness, and wellness [2]. One of the recent applications of pose estimation is for yoga pose detection. Yoga is an ancient practice that originated in South Asia and is gaining popularity worldwide. Yoga has been found to be helpful for preventative as well as diagnostic care across physiological and mental health [3,4]. Yoga practice not only improves flexibility, strength, and mindfulness but also provides improvements in patients with hypertension and cardiopulmonary and musculoskeletal ailments [5]. Due to the COVID-19 pandemic, yoga has become increasingly popular, as people are encouraged to stay indoors and practice at-home exercises. Yoga does not require specialized equipment or a lot of space, making it a convenient form of exercise. Yoga apps offer users the ability to customize their practice by selecting the level of difficulty, duration, and focus areas. This allows practitioners to achieve specific goals and needs. Additionally, yoga apps are a cost-effective alternative to in-person classes, making it easier for frequent practitioners to maintain their practice. Many yoga apps offer tracking and goal-setting features, which are helpful in staying motivated and seeing progress over time.

Correct posture alignment is crucial in the practice of yoga. Posture is essential for all forms of exercise, but it is particularly critical in yoga to obtain the benefits and prevent yoga-related sports injuries. Yoga encompasses a range of practices, including yoga poses known asanas, breathing and mindfulness exercises (pranayama and yoga nidra), and meditation. Yoga’s popularity has grown due to its potential positive outcomes in terms of physical performance and well-being [6,7]. As we live in a fast-paced society, yoga has become an increasingly popular form of exercise. Many people find yoga to be a valuable retreat from their hectic and busy lifestyles. Yoga, which comes in many forms, includes the popular style of Hatha, a blend of various styles that emphasizes physical exercises over still, meditative practices. The focus of Hatha yoga is on Pranayama, or breathing exercises, which are closely related to breathing biofeedback using auditory cues [8]. Yoga is primarily practiced for physical and mental well-being. According to their needs, people practice different kinds of yoga, such as prenatal yoga, baby yoga, children’s yoga, hot yoga, couple yoga, and seniors’ yoga. Yoga postures are commonly performed in all different practices. Many studies have demonstrated that yoga improves and optimizes the sports performance of athletes [9,10]. Activities that increase athletic performance are critical to maximizing training opportunities. As sport is a multidimensional endeavour, athletes may consider taking part in activities that optimize the specific dimensions of fitness as well as the aspects of performance in multiple dimensions [6].

There are several clinical studies demonstrating that home yoga practice with recommended yoga exercises lowers blood pressure and improves the quality of life in hypertensive patients [11]. For yoga trainers, health coaches, and yoga studios and students, the demand for technical assistance increased dramatically during the COVID-19 pandemic. Research analysis has found that AI helps practitioners and coaches make more personalized recommendations with different results and guidance on exercise accuracy [12]. The study found that exercise apps increased fitness and health activity by 27%, with improved health outcomes and higher levels of exercise. However, it is challenging to practice yoga unsupervised at home. There is also the possibility of yoga-related injuries. Therefore, it is imperative to develop solutions that can facilitate people practicing yoga at home, such as by using apps like BreathHRV (https://breathhrv.com/, accessed on 1 November 2023). As a precursor to building yoga apps for unsupervised pose detection, state-of-the-art machine learning and computer vision techniques are utilized, and the findings are presented in this paper. Several existing mobile applications have pioneered this field, utilizing technologies ranging from simple image processing to complex deep learning algorithms for pose estimation and feedback. These applications use a variety of techniques to identify and evaluate yoga poses, including convolutional neural networks (CNNs), keypoint detection, and pose estimation models such as OpenPose and PoseNet [13,14]. Garg et al. [15] explored a CNN- and MediaPipe-inspired deep learning (DL) approach for real-world yoga pose classification, demonstrating the effectiveness of advanced computer vision techniques in real-time applications. Similarly, Chasmai et al. [16] developed a view-independent classification framework for yoga postures, which aligns with the goal of providing accurate pose recognition regardless of the user’s position relative to the camera. Another study focused on classifying sun salutation yoga poses using machine learning techniques, achieving a remarkable accuracy of 96% with the KNN model. This research highlighted the potential of machine learning models in yoga pose classification when combined with pose estimation algorithms [17]. Additionally, a deep learning-based approach for yoga pose classification was presented at the 2022 International Conference on Machine Learning, Big Data, Cloud and Parallel Computing, underscoring the applicability of deep neural networks in this domain [18]. Our work builds on these foundational tools, with the aim of improving pose detection accuracy in a low-power device such as a smartphone for yoga practitioners.

## 2. Related Works

### 2.1. Importance of Yoga Apps

The primary reason for practicing yoga is to improve mental and physical health. Different age groups tend to choose various yoga practices, with the most common reason for practicing yoga being to enhance physical health. When selecting yoga apps to guide their practice, users are often most concerned with the user interface and ease of use. An overview of yoga’s possible benefits, as well as a framework for understanding them, can be seen in previous works [19]. The key to encouraging people to use yoga apps is to provide a better experience and a higher level of satisfaction. In one study, the Daily Yoga app was used as an object of investigation to study key factors related to the design of user experience and user satisfaction [20]. Yoga apps can also create a connection between the practitioner and their community. Table 1 presents some available yoga pose classification or detection Apps. Yoga practitioners can access downloadable apps from anywhere at any time on any device. Some apps focusing on heart rate variability [21] emphasize the breathing aspect of yoga. While yoga apps help connect yoga communities, there are challenges encountered, such as incorrect performance leading to injuries. Alternatively, human pose estimation can be achieved through computer vision techniques, with the end goal of deploying yoga pose estimation in Android and iOS applications. Yoga pose classification uses computer vision for several reasons. One of the main motivations is the personalization and automation of yoga instruction. Computer vision systems can provide feedback on posture and alignment as well as suggest modifications or alternative poses based on an individual’s performance of a yoga pose. More individualized instruction can be especially useful for beginners or those with injuries or other physical limitations. In this context, the authors of Chiddarwar et al. [22] showed that deep learning methods have proven to be extremely useful for the estimation of poses. According to their conclusions, PoseNet is the most suitable technique for implementing mobile applications, specifically for yoga. In their implementation of yoga pose classification, the authors of Long et al. [23] proposed the development of a yoga posture coaching system using an interactive display based on a transfer learning technique. Images were collected from eight volunteers (six women and two men), including yoga novices. The dataset covered 14 different postures, such as the bridge, cat-cow, child, cobra, corpse, downward-facing dog, sitting, extended side angle, warrior II, and warrior I postures. They used six transfer learning models (TL-VGG16-DA, TL-VGG19-DA, TL-MobileNet-DA, TL-MobileNetV2-DA, TL-InceptionV3-DA, and TL-DenseNet201-DA) for classification tasks to select the optimal model for the yoga coaching system based on evaluation metrics. They found the TL-MobileNet-DA model to be the optimal model, showing an overall accuracy of 98.43%, a sensitivity of 98.30%, and a specificity of 99.88%. In another study, an automated system was developed to identify and correct yoga poses which uses human action recognition (HAR) techniques [24]. The dataset included 18 yoga asanas, extending to 31 classes due to multi-facing views. The deep learning models trained included four modified and trained models: Xception, ResNet-50, VGG16, and VGG19.

Using deep learning techniques such as convolutional neural networks (CNNs) and transfer learning, the system recognizes yoga poses in real time and corrects the user with OpenPose, a multi-person 2D pose estimation algorithm. Users can practice 18 different asanas or poses with the system. Since users can face the camera from a left-hand side view, a right-hand side view, or a front view while practicing yoga, the system can predict asanas with 87.6% accuracy. Computer vision techniques have been applied to yoga pose classification in several studies, as mentioned earlier. In another study [25], a yoga pose correction system was implemented by training a model for only five distinct yoga poses using deep learning. According to the study, the deep learning model could identify these five poses with an accuracy of 95%. Although these methods are useful in yoga pose classification, they also require significant computational power, which is often lacking in smartphones, mostly when neural network models like CNNs are used. As per [26], the main challenges of deploying deep learning on mobile devices are limited computational resources such as memory and processing power, power constraints, limited storage, privacy and security concerns, and network connectivity. One study [27] stated that mobile devices often do not have much computational power, and thus it is difficult to run complex deep learning models. Power consumption is also a major problem.

### 2.2. Pose Estimation Methods

Yoga apps were briefly discussed, and we touched on pose estimation methods in the previous section (Section 1). The purpose of this section is to provide a brief overview of pose estimation as well as the difficulties associated with it. Yoga pose classification has been accomplished using a variety of computer vision techniques. Deep learning is a type of machine learning based on artificial neural networks in which multiple layers of processing are used to extract progressively higher-level features from data. By learning the patterns and features in the images, the neural network can classify new images based on those patterns and features [28]. Another approach is to use pose estimation techniques, which determine poses using estimates of key body points (like body joints). Methods like template matching or skeletal tracking can accomplish this by comparing the key points’ positions with predefined templates or models. It is also possible to combine these techniques, such as using deep learning to identify overall patterns in the images and using pose estimation to refine them. PoseNet is one such deep learning framework that is used for human pose identification in both image and video sequences. PoseNet has a sequence of fully connected layers. The architecture of PoseNet, based on the work in [29], is briefly described. A high-level architecture of PoseNet is discussed in [30]. The first component consists of an encoder that generates the encoding vector, which is an encoded representation of the features of the input images. The second component is the localizer, which generates another vector that denotes localization features. The last component is a regressor that is used to regress the final pose. In another work, the authors of [31] presented a real-time approach to detecting the 2D poses of multiple people in an image. In their method, body parts are associated with individuals using a non-parametric representation. As a result of their work, OpenPose [32] was released, the first open-source 2D pose detection system for multiple persons. Another work considered the task of estimating articulated human poses from real-world images. Based on CNN-based part detectors, the authors proposed a partitioning and labeling formulation of body part hypotheses. Their experiments on four different datasets showed state-of-the-art results for both single-person and multi-person pose estimation [33].

## 3. Proposed Approach

### 3.1. Dataset and Preprocessing

For the study, the Yoga-82 dataset developed in [34] containing 82 yoga pose classes with images downloaded from the web, including various camera angles, was used with MLKit Pose Detection to detect a subject’s body pose in real time from continuous videos or static images, utilizing skeletal landmark points for calculating the pose angles.

### 3.2. MLKit Pose Estimation

Estimating the pose of a human in 3D given an image or a video has found its applications in a range of applications, including human–robot interaction, gaming, sports performance analysis, and yoga or exercise pose estimation. While pose estimation presents its own set of challenges, various methods have been explored by researchers to tackle it, including the estimation of poses from single RGB images or sequences of such images. This work uses MLKit, which is a mobile software development kit (SDK) from Google that provides APIs for machine learning tasks. One of the tasks that MLKit can perform is pose estimation, which involves estimating the orientation and location of an object or a person in an image or video. MLKit uses the camera of the device and computer vision algorithms to analyze the image or video and provide an estimate of the pose.

### 3.3. Calculating the Angles

Yoga poses can be identified by computing the angles of various joints. These pose landmarks are used to compute the angles. Once all the angles are calculated which are needed to identify the pose, it can be checked to see if there’s a match, indicating that the pose has been identified. The X and Y coordinates from MLKit were used to calculate the angle between two body parts. This approach to classification has some limitations, as when only checking X and Y, the calculated angles vary according to the angle between the subject and the camera. Using the z coordinates can help overcome this problem. This is one of the limitations of our work.

The angles calculated were stored as features, and they are described in Table 2. These angles were further labeled and used for training and testing the classifiers for yoga pose discrimination. The flow chart explaining the process is provided in Figure 1. The pseudocode for a knee angle is shown below.

The Pseudocode is as follows (Figure 2):

hip_joint: a 2D vector representing the position of the hip;knee_joint: a 2D vector representing the position of the knee;ankle_joint: a 2D vector representing the position of the ankle;Outputs, namely angle_degrees: the angle in degrees between the vectors of the hip and knee as well as the knee and ankle.

The steps are as follows:

Compute the vector differences and magnitudes:
$$\begin{array}{cc} hip\_knee\_vector & =knee-hip\\ knee\_ankle\_vector & =ankle-knee\\ hip\_knee\_mag & =\mathrm{np}.\mathrm{linalg}.\mathrm{norm}\;(hip\_knee\_vector)\\ knee\_ankle\_mag & =\mathrm{np}.\mathrm{linalg}.\mathrm{norm}\;(knee\_ankle\_vector) \end{array}$$Compute the dot product and angle:
$$\begin{array}{cc} dot\_product & =\mathrm{np}.\mathrm{dot}(hip\_knee\_vector,knee\_ankle\_vector)\\ cosine\_angle & =dot\_product/(hip\_knee\_mag*knee\_ankle\_mag)\\ angle\_radians & =\mathrm{np}.\arccos\;(cosine\_angle)\\ angle\_degrees & =\mathrm{np}.\mathrm{degrees}\;(angle\_radians) \end{array}$$Return angle_degrees as the output.

### 3.4. Supervised Modeling

In this work, a machine learning framework is explored that harnesses feature angles calculated from the keypoint extraction of the human joint locations using MLKit pose estimation. All four broad paradigms of supervised classification were harnessed. The results are presented using curve-based methods such as traditional logistic and polynomial logistic regression, kernel-based methods such as support vector machines, tree-based methods such as boosted and bagged trees, as well as neural networks. In contrast to other approaches that are trained directly on the raw images, our approach helps provide greater transparency for the pose prediction problem. It was found that the extremely randomized tree (ERT) algorithm outperformed the others. The detailed discussion can be found in the subsequent sections. ERT algorithms employ an ensemble of decision trees, wherein a node split is chosen entirely at random, considering both the variable index and the variable splitting value. The fundamental concept behind ERT algorithms involves utilizing numerous small decision trees, with each serving as a weak learner. However, when combined in an ensemble, they collectively form a highly robust learner. ERT algorithms share similarities with other tree-based ensemble algorithms like random forests (RFs). Nevertheless, unlike RFs, ERT algorithms utilize the same training set for training all trees. Additionally, ERT algorithms differ from RFs in that they split a node based on both the variable index and variable splitting value, whereas random forests only split according to the variable value. This distinction contributes to ERT algorithms being more computationally efficient than RFs and possessing greater generalizability. In our study, we utilized an ensemble of 100 trees within the Extremely Randomized Trees (ERT) model. Each individual tree within this ensemble determined its splits based on the Gini coefficient.

## 4. Experiments and Results

In this section, the experimental set-up and results of our study are presented, which demonstrate the effectiveness of our proposed method on the benchmark Yoga-82 dataset.

### 4.1. Experiments

The original Yoga-82 dataset has 82 yoga poses [34] hierarchically arranged across three levels: pose name, pose class, and pose subclass, consisting of 6, 20, and 82 labels aggregated at each level, respectively. The authors used benchmark state-of-the-art CNN architectures including the DenseNet architecture for experiments using the Yoga-82 dataset. However, fog computing frameworks and mobile devices cannot easily deploy ultra-deep learning algorithms such as DenseNet due to their resource consumption. In [35], the authors showed that unsupervised clustering analytics might be feasible in fog frameworks. Considering such issues, shallow learning methods were explored for yoga pose classification rather than using deep learning, with the end goal of deploying the model in mobile devices with a small cloud computing resource cost and with a quick response time. In all, there were 1787 training images and 767 images for testing.

The models were trained and scored on a MacBook Pro (MacOS version 13.2.1) using Jupyter notebooks with a Python 3.9 kernel with the following specifications: a 2.7 GHz Quad-Core Intel Core i7 processor. Table 3 and Table 4 show the precision, recall, F1, and accuracy scores for cross-validation and on the test set, respectively.

In order to identify the most appropriate algorithm for our specific problem and dataset, different models were compared. Depending on the data and underlying relationships between variables, each machine learning algorithm has its own strengths and weaknesses. When comparing multiple models, we selected the one that provided the best balance of accuracy, generalizability, and interpretability for our particular use case, ultimately leading to more reliable and robust predictions. Yoga pose classification was evaluated using classification metrics and a confusion matrix.

Figure 3 shows the confusion matrix for the cross-validation and test results. Summary plots are used for the Shapley [36] values for each feature for feature importance. Shapley values are a concept borrowed from cooperative game theory for machine learning. The Shapley value gives us the average marginal contribution of a feature value to the final outcome. The Shapely plots in Figure 4 demonstrate the average contribution of each predictor to the final outcome for a given class as well as the overall model predictions. For instance, in the extremely randomized trees model, the LKA was the most predictive angle for class 4, but the RKA was the most predictive angle overall across all classes. It was also observed that the elbow angle features had the lowest contribution to the model performance uniformly across different sets of algorithms from kernel-based algorithms to tree-based algorithms to neural networks.

### 4.2. Results

In the cross-validation experiments, extremely randomized trees (ERT) had 91% prediction accuracy. Not only did it achieve the best accuracy results for the ERTs, but it also performed well across all classes, including the minority classes. Logistic regression had the lowest prediction accuracy at 69%, but polynomial logistic regression improved the accuracy to 85%. SVMs did not improve the accuracy (84%). On the holdout test set, the performance for random forests was 89%, while it was 88% for gradient boosting, 89% for extreme gradient boosting, and 88% for deep neural networks. Similarly, for the test dataset, extra trees had 92% for its prediction accuracy. Logistic regression had the lowest prediction accuracy at 67%, but polynomial regression improved the accuracy to 83%. The prediction accuracy for random forests was 90%, while it was 89% for gradient boosting, 90% for extreme gradient boosting, and 85% for deep neural networks. For deep neural networks, the prediction accuracy dropped for the test dataset compared with the cross-validation dataset. In summary, extra trees outperformed the other algorithms on both the cross-validation and test sets and generalized the best among them. The Table 5 shows the accuracy comparison. This comparison demonstrates the accuracy of our architecture in relation to other architectures, highlighting the differences in performance.

Table 6 demonstrates the prediction latency averages after running each model for 200 runs. In Figure 5, it can be observed that the test accuracy improved as the number of trees increased to 100. Although the best extra trees model used 100 trees, it can be observed that even with 25 trees, the extra trees models had an accuracy higher than those of the other models. With even as few as 10 trees, the extra trees models reached a similar performance to the next best model using random forests with 100 trees. For low-latency devices, a lower number of trees can be chosen, which can reduce prediction latency by three times with minimal loss to the accuracy.

### 4.3. Limitations

Table 5 provides a comparison of the accuracies, specifically focusing on the Top-1 accuracy, a standard metric in model evaluation. This metric necessitates that the model’s most probable prediction (the one with the highest probability) must align precisely with the expected correct answer. This approach was adopted to gauge the performance of our model against others. It is important to acknowledge a key limitation in our study: our utilization of only the first level of the Yoga-82 dataset’s hierarchical structure for model training and evaluation. The Yoga-82 dataset is organized into a three-tier hierarchy, with the third level being the most detailed one, encompassing 82 distinct classes. Our analysis, however, concentrated on the first level, which comprises six broader categories (balancing, inverted, reclining, sitting, standing, and wheel). This approach contrasts with the deep learning architectures it was compared against, which employ the more granular third level. Looking ahead, our future work aims to expand our research to incorporate more intricate classes, including but not limited to the third level of the Yoga-82 dataset. This expansion will enable a more detailed and nuanced analysis, potentially enhancing the model’s applicability and accuracy. Despite these limitations, this approach, centering on angle-based feature calculation combined with a shallow machine learning framework, holds significant promise, especially for applications in low-latency devices like smartphones. This method could pave the way for efficient, real-time pose detection, a crucial feature for a wide array of practical applications in the field of mobile computing and beyond.

## 5. Conclusions

This paper has demonstrated promising results for yoga pose classification using machine learning and computer vision techniques, taking the first steps toward creating a yoga pose classification app that is not only accurate but also suitable for Android devices with low computational power. The primary aim of our proposed method is to enable accurate and efficient yoga pose classification, minimizing the possibility of injury and promoting unsupervised training, especially during times like the COVID pandemic, when in-person guidance may be limited. Our objectives included (1) leveraging state-of-the-art human pose estimation techniques to manually extract relevant features for improving classification accuracy, (2) exploring new possibilities for yoga pose classification by applying the extracted features, and (3) overcoming the limitations of resource-intensive methods, such as convolutional neural networks, by devising novel approaches that achieve state-of-the-art accuracy using manually selected features. In addition to its health benefits, yoga is also used for spiritual and mental well-being, and our proposed method has the potential to enhance these aspects by providing accurate and accessible yoga pose guidance through an app-based platform. Among the ML models, randomized trees outperformed all other models, with 91% prediction accuracy for the test dataset and 92% accuracy for the cross-validation dataset. In six yoga classes (balancing, inverted, reclining, sitting, standing, and wheel), the classification accuracy has been promising. Nevertheless, yoga asanas are infinite, and developing models for each of them is incredibly time-consuming. Although our current study focused on extracting yoga pose features manually from images, it will also work with real-time video captured by a phone camera using Machine Learning Kit (MLKit) pose estimation. In the realm of yoga, ample opportunities exist for further research and development in posture detection. Expanding the application’s capability to recognize a broader range of poses can accommodate diverse yoga styles and practices. Integrating supplementary data sources, such as motion capture or wearable sensors, may yield more precise feedback on posture and alignment. Combining pose estimation with biofeedback holds great promise for enhancing the user experience. Tailoring applications to suit specific populations, including beginners and individuals with disabilities, could broaden their utility. The methodology presented in this paper for detecting yoga asanas demonstrated markedly superior accuracy, offering a valuable foundation for future yoga app development and opening doors to new research avenues. Future research in the realm of technology-assisted yoga practice could focus on several key areas. This includes the development of advanced 3D pose estimation techniques, the integration of biometric feedback to enhance personalization, and machine learning algorithms for tailored yoga experiences.

## Figures and Tables

**Figure 1 healthcare-11-03133-f001:**
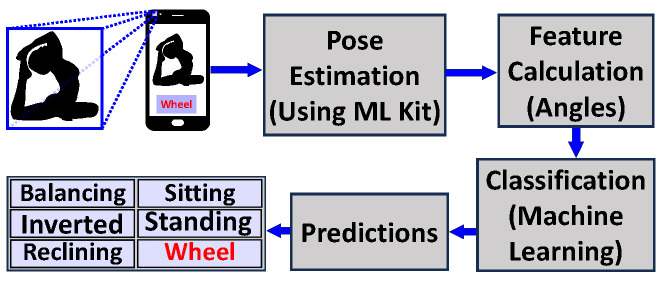
Flowchart illustrating the process of using machine learning for the classification of poses, using images as input. The images are first processed using MLKit to extract features, specifically the angles between body parts. These features are then input into a machine learning classifier, which is trained to predict the pose of the individual in the image.

**Figure 2 healthcare-11-03133-f002:**
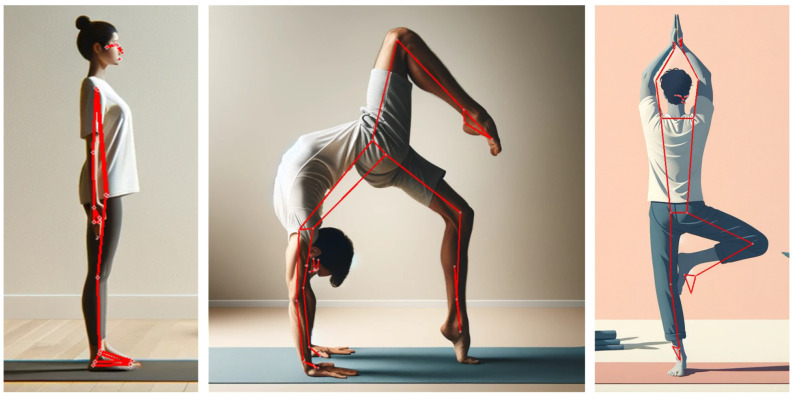
Pose detection using MLKit. Red lines connect the body joints on three different test images created using ChatGPT.

**Figure 3 healthcare-11-03133-f003:**
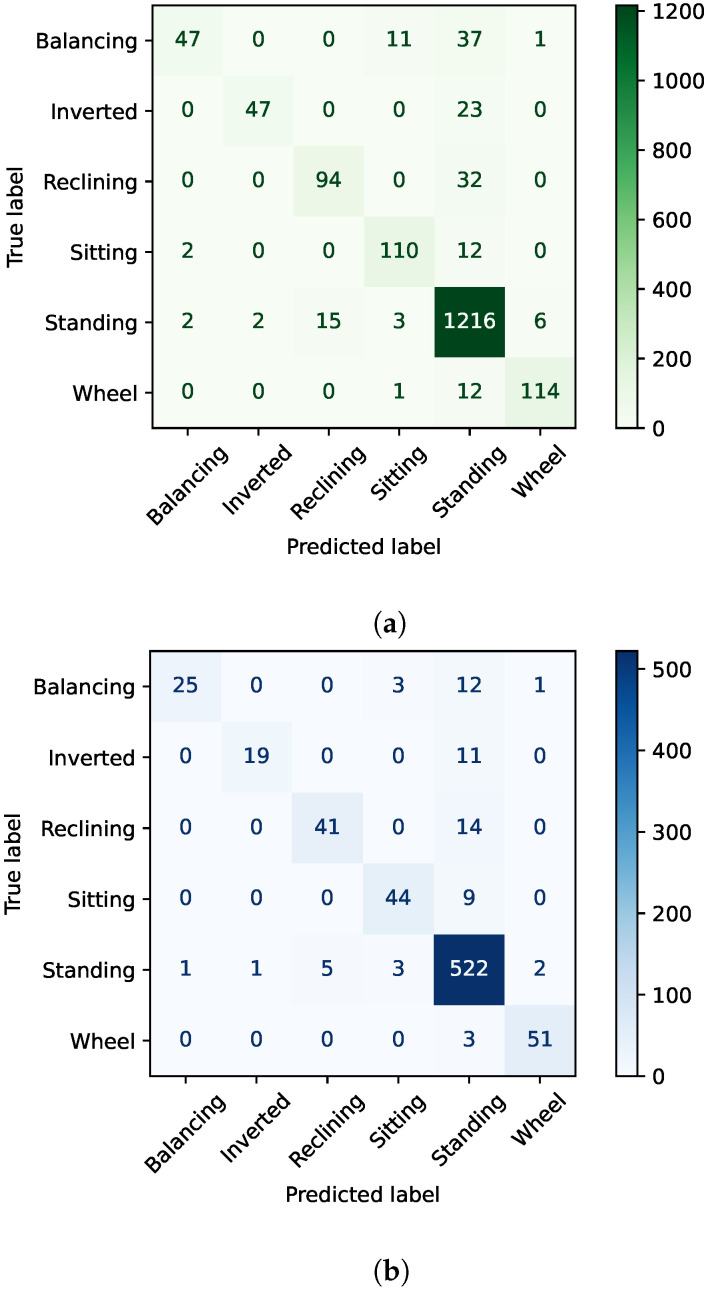
(**a**) Cross-validation results. (**b**) Holdout test set results. Confusion matrices are for the best performing algorithm (extremely randomized trees) across the cross-validation and test set results.

**Figure 4 healthcare-11-03133-f004:**
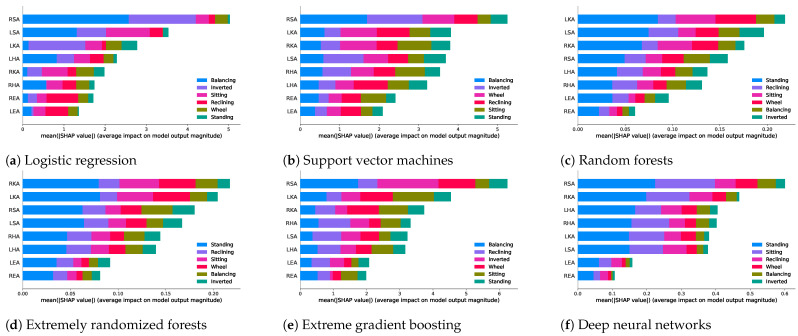
Shapley feature importance plots on the test dataset across different explored algorithms. Each subplot demonstrates the average impact of a given feature in predicting a given label aggregated over the entire dataset.

**Figure 5 healthcare-11-03133-f005:**
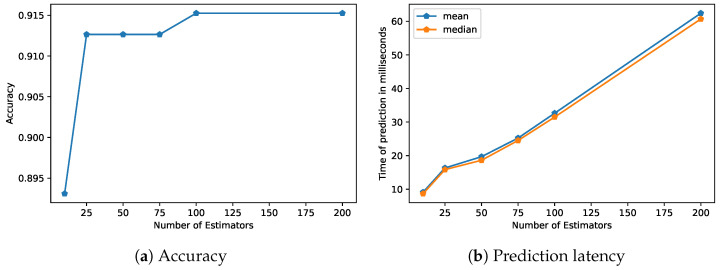
Latency performance analysis for the extremely randomized trees (extra trees) model. This figure presents a summary of the latency experiments conducted to assess the real-time performance of the extra trees model.

**Table 1 healthcare-11-03133-t001:** Overview of yoga apps.

App Name	Origin	Features
FitYoga	India	Offers guided yoga and meditation with AI-driven real-time movement tracking for posture improvement. Features include chakra balancing and wellness practices. https://fityoga.app/, accessed on 1 November 2023
Sofia	India	AI-based yoga training with Google sign in. Features voice-assisted Asanas, real-time pose recognition, and rewards for pose accuracy. https://soumi7.github.io/project-sofia.html, accessed on 1 November 2023
YogAI	UK	Features a learning and practice module with TensorFlow PoseNet and ml5js for pose detection and classification. Requires camera access for pose matching. https://cris-maillo.github.io/yogAI/index.html, accessed on 1 November 2023.
YogaIntelliJ	India	A web-based yoga application that predicts body coordinates for yoga poses using a webcam and provides feedback on pose accuracy. https://eager-bardeen-e9f94f.netlify.app/, accessed on 1 Decemeber 2023
Yoga Studio by Gaiam	USA	Offers over 180 classes and custom flows. Includes meditations but requires a subscription for some features. Layout can be complex, and updates may cause glitches. https://yogastudioapp.com/gaiam, accessed on 1 Decemebre 2023
Down Dog	USA	Customizable practices with various styles and instructor voices. Focuses on beginner to intermediate levels but requires a membership for full access and offers limited music in the free version. https://www.downdogapp.com/, accessed on 1 November 2023
Daily Yoga	China	Features the largest global yoga community with new classes weekly. Offers extensive yoga and meditation classes but is more expensive and advanced, requiring a subscription for full access. https://www.dailyyoga.com/, accessed on 1 November 2023

**Table 2 healthcare-11-03133-t002:** Description of angle-based features and their descriptions.

Feature	Description	Feature	Description
LSA	Left Shoulder Angle	RSA	Right Shoulder Angle
LEA	Left Elbow Angle	REA	Right Elbow Angle
LHA	Left Hip Angle	RHA	Right Hip Angle
LKA	Left Knee Angle	RKA	Right Knee Angle

**Table 3 healthcare-11-03133-t003:** Summary of the classification metrics for the fivefold cross-validation experiments across the different algorithms. To emphasize their high values, the Extra Trees are distinguished in bold formatting.

Algorithm	Precision		Recall		F1		Accuracy
	**(Overall)**	**(Weighted)**	**(Overall)**	**(Weighted)**	**(Overall)**	**(Weighted)**	
Traditional Logistic Regression	0.27	0.56	0.26	0.69	0.24	0.61	0.69
Polynomial Logistic Regression	0.78	0.84	0.67	0.85	0.72	0.84	0.85
Support Vector Machines	0.84	0.84	0.60	0.84	0.67	0.82	0.84
Random Forest	0.90	0.89	0.73	0.89	0.79	0.88	0.89
**Extremely Randomized Trees (Extra Trees)**	**0.91**	**0.91**	**0.78**	**0.91**	**0.83**	**0.91**	**0.91**
Gradient Boosting	0.86	0.88	0.72	0.88	0.77	0.87	0.88
Extreme Gradient Boosting (XGBoost)	0.86	0.89	0.75	0.89	0.80	0.89	0.89
Deep Neural Networks	0.83	0.87	0.73	0.88	0.77	0.87	0.88

**Table 4 healthcare-11-03133-t004:** Summary of the classification metrics on the test set across the different algorithms. To emphasize their high values, the Extra Trees are distinguished in bold formatting.

Architecture	Precision		Recall		F1		Accuracy
	**(Overall)**	**(Weighted)**	**(Overall)**	**(Weighted)**	**(Overall)**	**(Weighted)**	
Traditional Logistic Regression	0.27	0.56	0.24	0.67	0.21	0.59	0.67
Polynomial Logistic Regression	0.76	0.82	0.65	0.83	0.69	0.82	0.83
Support Vector Machines	0.82	0.83	0.60	0.83	0.66	0.81	0.83
Random Forest	0.91	0.90	0.74	0.90	0.79	0.89	0.90
**Extremely Randomized Trees (Extra Trees)**	**0.92**	**0.92**	**0.79**	**0.92**	**0.84**	**0.91**	**0.92**
Gradient Boosting	0.89	0.89	0.74	0.89	0.80	0.89	0.89
Extreme Gradient Boosting (XGBoost)	0.88	0.89	0.74	0.90	0.80	0.89	0.90
Deep Neural Networks	0.84	0.85	0.65	0.85	0.71	0.83	0.85

**Table 5 healthcare-11-03133-t005:** Comparison of Top-1 accuracy across different architectures on the Yoga-82 dataset. Note: previous studies used level III poses of the Yoga-82 dataset. The current study used level I poses. More discussion is in the Limitations section.

Architecture	Top-1 % Accuracy
ResNet-50	63.44
ResNet-101	65.84
ResNet-50-V2	62.56
ResNet-101-V2	61.81
DenseNet-121	73.48
DenseNet-169	74.73
DenseNet-201	74.91
MobileNet	67.55
MobileNet-V2	71.11
ResNext-50	68.45
ResNext-101	65.24
**Extremely Randomized Trees (Extra Trees)**	**92**

**Table 6 healthcare-11-03133-t006:** Summary statistics of prediction latency over 200 runs for each algorithm on the test set in milliseconds.

Algorithm	Prediction Latency	
	**(Mean)**	**(Median)**
Logistic Regression	2.587	2.390
Polynomial Regression	2.129	1.490
Support Vector Machines	24.327	24.115
Random Forest	33.698	32.593
Extremely Randomized Trees (ExtraTrees)	33.827	31.371
Gradient Boosting	13.189	12.290
Extreme Gradient Boosting (XGBoost)	17.060	16.724
Deep Neural Networks	64.200	59.813

## Data Availability

Upon request, the data can be made available.

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
