# Peer review of "Yoga Pose Estimation Using Angle-Based Feature Extraction"

_healthcare, 2023, doi:10.3390/healthcare11243133_

Round 1

Reviewer 1 Report

Comments and Suggestions for Authors

The paper is well-written, focusing on an emergent topic in the field of computer vision and ML.

1.      Abstract

·        Line 8: Please add that models were first trained and then tested.

·        Line 10: 92% percent cross validation: What do you mean by cross validation here?

·        The abstract should contain some of the information about the dataset.

·        Also add what was the missing in the previous studies and why the current study has been conducted.

·        The limitation of the study needs to be added in the abstract.

2.      Introduction:

Line 34: Add the reference.

Line 63-63: Line 63-64 is the repetition of line 35-36

Line 69-70  is the repetition of line 43-44

3.      Related works:

Line 142-143, 138-139 remove the titles of the study

Line 148, there is no section A

The table need to be added in the related work to represent the summary of the yoga apps, their features and the results. Furthermore, in the literature review, I couldn’t find any such thing related to the dataset. We need to know that which datasets has been used and also the details about the dataset. Because if the data set is imbalance, then we can’t rely on the accuracy of the model. Therefore, better to add the dataset details related to each study.

4.      Proposed Approach

The details need to be added related to the dataset like the number of total images and the number of images per class. Also, the date when the dataset was released and the motivation why the specific dataset has been selected.

The description of the ML models is missing in the study. The parameters used is missing.

5.      Experiments and Results 

Figure 4, description need to be added in the discussion.

The benchmark table need to be added to compare the results of the previous studies that was performed using the same dataset. Also, the limitation of those studies needs to be specified and how those limitations were addressed in the current study.

The limitation of the current study needs to be added. Also, future work needs to be added in the conclusion. The author needs to add the contribution section that enlist the main contribution of the study.

Most of the references in the study are old latest studies in 2022 and 2023 need to be added.

Comments on the Quality of English Language

The paper just need minor modification related to language. Overall it is good interms of language. 

Reviewer 2 Report

Comments and Suggestions for Authors

Thank you very much for the opportunity to review your work. There are several aspects that should be improved. I'll try to list the suggestions below:

Rewrite the tile. It would be better something like: “Yoga Pose Estimation using Angle-Based Feature Extraction”, I think there is no need for “Machine Learning”.

Line 28 - “Pose estimation has become essential in the field of sports, fitness and wellness.”. Examples of this in this field? Include citations to it.

Line 32 – “Yoga practice not only improves flexibility, strength, and mindfulness but also provides improvements in patients with hypertension, cardiopulmonary and musculoskeletal ailments.” Again, you should also reference something to support this.

Line 49 - “Yoga’s popularity has grown due to its potential positive outcomes in physical performance and well-being [2,3].”, maybe use other references instead of repeating the ones from 2 paragraphs before?

There are a lot of phrases, mainly in the introduction that are too short that can be integrated in the previous or after sentence. A sequence of phrases that start in line 52, are a good example of this: “Yoga comes in many forms. A popular style of yoga is Hatha (a combination of many styles). As opposed to still, meditative yoga, it is more physical. Pranayama (breathing exercises) are the focus of Hatha yoga. A close relative of this is breathing biofeedback using auditory cues [4]. Yoga is primarily practised for physical and mental well-being.”

Line 59 – “Many studies have demonstrated that yoga improves and optimises the sports performance of athletes [5].” Although many studies are refered, only one is cited...

Line 69 – “The results of the survey showed that yoga and fitness apps have become an integral part of the daily routines of practitioners because they allow them to practice at any time at their convenience.” The survey is not mentioned before?

Line 78 – “BreathYogi” – reference is missing?

Overall, the introduction needs to be revised. Take also into account that all references should be referenced like this (this is also applied to whole document): “lllll [x]” and not like this: “lllll[x]”. Also remember that “convolutional neural networks (CNN)”, should be: “Convolutional Neural Networks (CNN), revise it in the abstract as well.

Abbreviations should be provided for Deep Learning – DL, Machine Learning – ML, Human Action Recognition – HAR, etc.. You didn’t used this, although in line 298 you use the term ML, without saying what ML means. 

Section 2 also needs an grammar review and proof reading. 

Line 150 – “Deep learning is a type of machine learning that involves training a neural network on a large dataset of labelled images.” Although it is usually the situation, both are not connected. The definition of deep learning is not indexed to the training form.

After section 2 you forgot to use sections. Is this correct?

In “Section 3” another example of short sentences, that should be revised. “We have used the Yoga-82 dataset developed by[28] for our study. The dataset contains yoga pose images downloaded from the web. The images also included different camera view angles. There were 82 yoga pose classes in the dataset.”

Line 183 - “Estimating the pose of a human in 3D given an image or a video has found its applications in a range of applications”, has a redundancy

Also, you use “we” very often along the paper, which is not very formal and should not be a regular term in a scientific paper. Please, rewrite using passive.

One final remark is the objective of the paper... it starts by point the reader to the correctness of performing the exercises. Nevertheless, the approach followed is just the recognition of the pose being done, regardeless of being correctly done. This is contrary of the main objective...

Comments on the Quality of English Language

Overall, there are many telegraphic sentences (too short) that complicates the reading. Please, perform a proof reading to solve also some grammar issues.
